# The Effect of Meat and Bone Meal (MBM) on Phosphorus (P) Content and Uptake by Crops, and Soil Available P Balance in a Six-Year Field Experiment

**Aleksandra Załuszniewska * and Anna Nogalska** 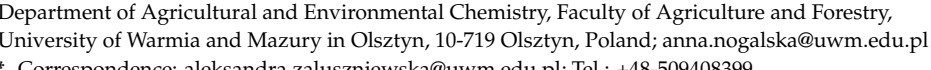

Department of Agricultural and Environmental Chemistry, Faculty of Agriculture and Forestry, University of Warmia and Mazury in Olsztyn, 10-719 Olsztyn, Poland; anna.nogalska@uwm.edu.pl
* Correspondence: aleksandra.zaluszniewska@uwm.edu.pl; Tel.: +48-509408399

**Abstract:** The aim of a six-year field experiment conducted in north-eastern (NE) Poland was to determine the effect of meat and bone meal (MBM) on phosphorus (P) content and uptake by different crops, soil available P balance, and soil pH. Five treatments were established: (1) zero-fert; (2) inorganic NPK; (3) $1.0 \, t \, ha^{-1}$ MBM; (4) $1.5 \, t \, ha^{-1}$ MBM; and (5) $2.0 \, t \, ha^{-1}$ MBM. Constant nitrogen (N) and potassium (K) rates and increasing P rates (0.0; 45; 68 and 90 kg $ha^{-1}$) were applied. The lowest dose of MBM, which supplied 45 kg P $ha^{-1}$ each year, was sufficient to meet the P requirements of silage maize, winter wheat, and winter oilseed rape to the same extent as mineral P fertilizer at the equivalent rate. The uptake, balance, and utilization of P by plants were comparable in both treatments. Phosphorus applied each year at high rates (68 and 90 kg $ha^{-1}$) with two higher MBM doses contributed to excessive P accumulation in soil; therefore, MBM should not be applied at doses exceeding $1.5 \, t \, ha^{-1}$ to crops grown in acidic soils. Soil pH was not significantly affected by MBM. MBM can replace conventional mineral P fertilizers in crop cultivation.

**Keywords:** maize; wheat; oilseed rape; soil; phosphorus balance; animal meal; phosphorus availability

## 1. Introduction

Phosphorus (P) is one of the three essential macronutrients required for the healthy growth and development of plants. Under current economic and technological conditions, natural phosphate rock reserves (the main source of conventional P fertilizers) are estimated at 18 billion tons and may be depleted within the next 50–100 years [1]. Therefore, cheaper and environmentally safe alternatives are being sought. One of the solutions is to use organic waste [2,3], including meat and bone meal (MBM) [4,5], as a source of nutrients for plants. Meat and bone meal is obtained during the processing of raw animal materials, and it includes slaughterhouse waste not intended for human consumption such as heads, hooves, blood, fat, feathers, bones, and giblets. These by-products account for around 30% of the animal's live weight [1]. For many years, MBM had been used in the production of high-protein animal feeds; however, the European Union introduced a ban on the use of processed animal protein in cattle, poultry, and pig diets to control Bovine Spongiform Encephalopathy (BSE). Therefore, other methods of animal waste disposal had to be developed. Meat and bone meal can be used for agricultural purposes because it is rich in nutrients needed for plant growth, such as nitrogen (N), P, and calcium (Ca) as well as micronutrients and organic matter [6–9]. In comparison with manure, MBM contains four times more N, ten times more P, and eight times more Ca on a dry matter (DM) basis. However, it is less abundant in potassium (K) and magnesium (Mg). It is recommended to use MBM instead of, or in addition to, conventional mineral fertilization to improve P recycling. It can also replace manure in organic farms that do not keep livestock. The use of MBM in agriculture is environmentally justified because the recycling of nutrients from organic waste contributes to conserving non-renewable resources [5,10–12]. From a

sustainability point of view, there are strong arguments for nutrient recycling, including the use of organic nutrient sources in agriculture.

The research hypothesis postulates that MBM is a good source of P for crops; it increases the available P content of soil and stabilizes soil pH. The hypothesis was tested during a long-term field experiment involving the cultivation of the most economically important crop species. It was assumed that MBM can replace conventional mineral P fertilizers. The aim of this study was to evaluate the effect of MBM applied at three doses on P content and uptake by silage maize, winter wheat, and winter oilseed rape; the content and balance of available P in soil; and soil pH. This study is part of long-term research (a six-year field experiment) investigating the influence of MBM on yield components, the chemical composition of crops, and selected properties of soil. The results regarding the yield and quality of winter oilseed rape [13,14]; the yields and N content of silage maize, winter wheat, and oilseed rape; and soil N balance [15] have already been published.

## 2. Materials and Methods

### 2.1. Experimental Site

A long-term small-area field experiment with silage maize, winter wheat, and winter oilseed rape was conducted at the Agricultural Experiment Station in Tomaszkowo (NE Poland) owned by the University of Warmia and Mazury in Olsztyn, in 2014–2019. The experiment was established on brown soil developed from loamy sand, classified as Dystric Cambisol according to the World Reference Base (WRB) [16]. It had a randomized block design with four replications. The soil was slightly acidic (pH in 1 M KCl = 5.61), it had mineral N content of 8.82 mg $kg^{-1}$, and it was moderately abundant in available P (65 mg $kg^{-1}$), abundant in K (163 mg K $kg^{-1}$), and highly abundant in Mg (96 mg $kg^{-1}$ soil). Meat and bone meal used in the experiment had the following chemical composition, per kg DM: 963 g DM, 710 g organic matter, 280 g crude ash, 137 g crude fat, 78.7 g N, 45.3 g P, 3.32 g K, 100.1 g Ca, 6.8 g Na, and 2.0 g Mg $kg^{-1}$. Its pH in $H_2O$ was 6.3. The meat and bone meal was applied in a loose form and mixed with soil without additional processing. The MBM was low-risk (category 3) material obtained from SARIA Poland Ltd. (Sarval Plant in Długi Borek, NE Poland) [15].

### 2.2. Experimental Design

Five treatments were established: (1) without fertilization (zero-fert); (2) 158 kg N, 45 kg P and 145 kg K $ha^{-1}$ as mineral fertilizer (inorganic NPK); (3) 1.0 t $ha^{-1}$ MBM and 79 kg N $ha^{-1}$ as mineral fertilizer and 145 kg K $ha^{-1}$ as mineral fertilizer (1.0 t $ha^{-1}$ MBM); (4) 1.5 t $ha^{-1}$ MBM and 40 kg N $ha^{-1}$ as mineral fertilizer and 145 kg K $ha^{-1}$ as mineral fertilizer (1.5 t $ha^{-1}$ MBM); and (5) 2.0 t $ha^{-1}$ MBM and 145 kg K $ha^{-1}$ as mineral fertilizer (2.0 t $ha^{-1}$ MBM) (Table 1).

**Table 1.** Annual rates of nitrogen (N), phosphorus (P), and potassium (K) applied with meat and bone meal (MBM) and mineral fertilizers (kg $ha^{-1}$) to silage maize (2014 and 2015), winter wheat (2014/15 and 2017/18), and winter oilseed rape (2015/16 and 2016/17) [15].

| Treatment | N | P | N:P ***** | K |
|:---:|:---:|:---:|:---:|:---:|
| 1. Zero-fert | 0 | 0 | 0 | 0 |
| 2. Inorganic NPK * | 158 | 45 | 1:0.3 | 145 |
| 3. 1.0 t MBM + $N_{79}$ ** | 158$_{(79+79)}$ | 45 | 1:0.3 | 145 |
| 4. 1.5 t MBM + $N_{40}$ *** | 158$_{(118+40)}$ | 68 | 1:0.4 | 145 |
| 5. 2.0 t MBM **** | 158 | 90 | 1:0.6 | 145 |

* Inorganic NPK—mineral fertilization; ** MBM + $N_{79}$—meat and bone meal with mineral nitrogen (79 kg N $ha^{-1}$) fertilizers; *** MBM + $N_{40}$—meat and bone meal with mineral nitrogen (40 kg N $ha^{-1}$) fertilizers; **** MBM—meat and bone meal fertilizer. Every year, mineral potassium fertilizer was applied at the same rate of 145 kg K $ha^{-1}$ in treatments 2, 3, 4 and 5; ***** N:P—ratio of nitrogen to phosphorus in the fertilizer.

In order to achieve a constant level of N fertilization (158 kg $ha^{-1}$), MBM was supplemented with mineral N at 40 kg $ha^{-1}$ (treatment No. 3) and 79 kg $ha^{-1}$ (treatment No. 4) as urea (46% N) or ammonium nitrate (34% N), to widen the narrow N:P ratio in MBM

(1:0.6). The widest N:P ratio of 1:0.3 was noted in treatments No. 2 and 3, and it reached 1:0.4 in treatment No. 4, and 1:0.6 in treatment No. 5 (MBM only). Due to the low K content of MBM (approx. 3.3 g kg$^{-1}$ DM), K was applied before sowing in mineral form (potash salt, 49.8% K) at a constant rate (145 kg K ha$^{-1}$) in all treatments (No. 2–5). Constant N and K rates and increasing P rates (0.0, 45, 68, and 90 kg ha$^{-1}$) were applied. In the NPK treatment (No. 2), 45 kg P ha$^{-1}$ was applied as granular triple superphosphate (20.1% P). In treatments No. 3, 4, and 5, P was supplied only with MBM [15].

### 2.3. Crop Plants

Three crop species were grown in a six-year rotation system: silage maize (cv. PIONIER P8488), winter wheat (cv. ARKADIA), and winter oilseed rape (hybrid cv. SY SAVEO). Maize was grown in the first (sowing date: 5 May 2014; harvest date: 22 September 2014) and sixth (sowing date: 29 April 2019; harvest date 24 September 2019) year, winter wheat was grown in the second (sowing date: 25 September 2014; harvest date: 4 August 2015) and fifth (sowing date: 27 September 2017; harvest date: 23 July 2018) year, and winter oilseed rape was grown in the third (sowing date: 26 August 2015; harvest date: 18 July 2016) and fourth (sowing date: 25 August 2016; harvest date: 20 July 2017) year of the experiment. Winter rye was the forecrop of maize grown in the first year of the study. Each tested crop was grown twice in the six-year rotation system to compare the responses of each species to the experimental factor. The plants were grown on the same 20 plots, and the area of each plot was 20 m$^2$ (4 × 5 m). A 1 m guard row was maintained between the plots. All cultivation and crop protection measures were applied at the optimum dates, in accordance with good agricultural practices. The crops were harvested with a harvester from each plot. An average representative plant sample of 1.0 kg was taken from each plot. At harvest, in each plot, yield was determined in terms of weight after threshing, and moisture content was measured. All treatments were described in detail by Nogalska and Załuszniewska [15].

### 2.4. Chemical Composition of Plants

The 1 kg samples of maize herbage, winter wheat grain and straw, and winter oilseed rape seeds and straw were collected from each plot for chemical analyses. Plant samples, which had been dried to absolutely dry mass at 105 °C, weighted, and ground, were then wet mineralized in concentrated sulfuric (VI) acid with hydrogen peroxide (H$_2$O$_2$) as the oxidizing agent. Mineralized plant samples were analyzed for P content by the vanadium-molybdenum method (UV-1201 V spectrophotometer, Shimadzu Corporation Kyoto, Japan).

Phosphorus uptake (P$_P$) by the aboveground biomass of maize, winter wheat, and winter oilseed rape was calculated by multiplying the P content of herbage, grain/seeds, and straw by yields (on a DM basis).

The utilization of P (W$_P$) from P fertilizers and MBM was calculated using the following formula:

$$W_P = \frac{(P_P - P_0)}{P} \times 100\%$$

where:

W$_P$—coefficient of P utilization (%),
P$_P$—P uptake with herbage, grain, seeds, and straw in the P-fertilized treatment (kg·per ha),
P$_0$—P uptake with herbage, grain, seeds, and straw in the zero-fert treatment (kg·per ha),
P—P rate (kg·ha$^{-1}$).

### 2.5. Chemical Composition of Soil

Soil samples were collected before the experiment and each year after harvest to determine pH in 1 mol KCl dm$^{-3}$ (soil: solution extraction ratio 1:2.5) by the potentiometric method (pH-metr CP-505, Elmetron Sp. j. Zabrze, Poland). The available P was deter-

mined by the Egner-Riehm DL method (soil: solution extraction ratio 1:50) (UV-1201 V spectrophotometer, Shimadzu Corporation Kyoto, Japan).

### 2.6. Weather Conditions

The six-year experimental period (2014–2019) was characterized by varied weather conditions, in particular with regard to precipitation distribution (Figure 1). During the growing seasons, temperatures were higher by around 0.6 °C than the long-term average of 1981–2010 (Figure 2). High temperatures could support the growth of maize, which is a thermophilic species, and the release of nutrients from MBM, but uneven precipitation was not conducive to this process. A detailed description of weather conditions during the growing season of the three tested crops that were grown twice can be found in the work of Nogalska and Załuszniewska [15].

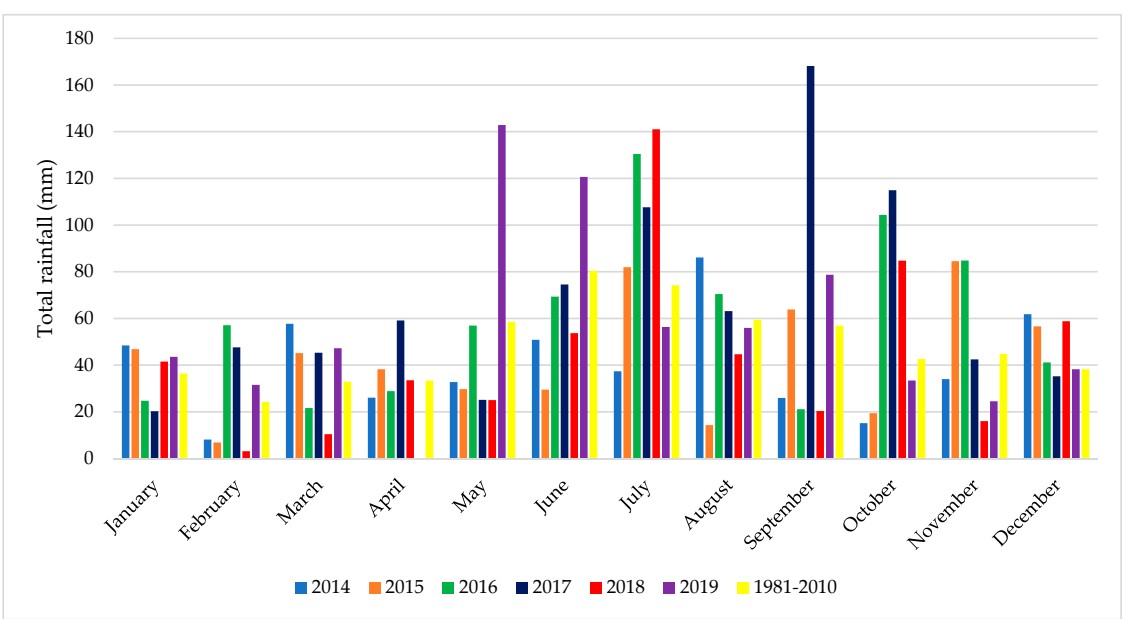

**Figure 1.** Total rainfall (mm) in 2014–2019, and in the 1981–2010 reference period according to the Research Station in Tomaszkowo.

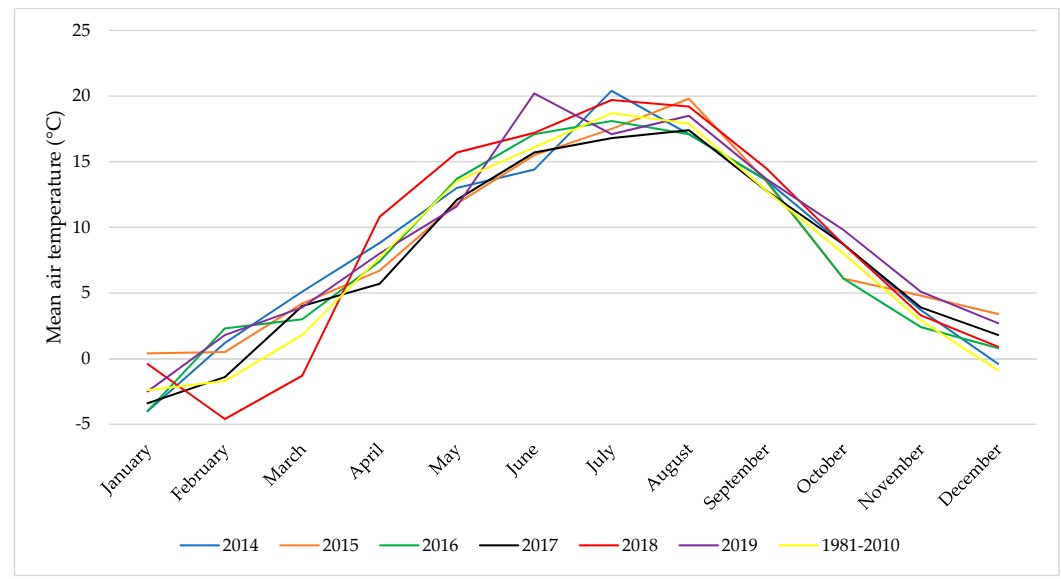

**Figure 2.** Mean air temperature (°C) in 2014–2019, and in the 1981–2010 reference period according to the Research Station in Tomaszkowo.

*2.7. Statistical Analysis*

Data on crops (P content of maize herbage, winter wheat grain and straw, and winter oilseed rape seeds and straw, and P uptake by aboveground biomass) were processed by one-way (five fertilization treatments) ANOVA in the STATISTICA 12 program [17]. Data on soil (P content and pH in 1 M KCl $dm^{-3}$) were processed by two-way repeated measures ANOVA in the STATISTICA 12 program. Increasing MBM dose was the grouping factor (five fertilization treatments), and year of the study was the repeated measurement factor (six years).

## 3. Results and Discussion

*3.1. Phosphorus Content of Crops*

In the six-year field experiment, three increasing doses of MBM led to significant changes in the P content of two tested crops—winter oilseed rape and winter wheat (Table 2).

**Table 2.** The effect of increasing MBM doses on the phosphorus content (g $kg^{-1}$ DM) of crops in rotation.

| Treatments | 2014 Silage Maize | 2014/15 Winter Wheat | | 2015/16 Winter Oilseed Rape | | 2016/17 Winter Oilseed Rape | | 2017/18 Winter Wheat | | 2019 Silage Maize |
|---|---|---|---|---|---|---|---|---|---|---|
| | | Grain | Straw | Seed | Straw | Seed | Straw | Grain | Straw | |
| 1. Zero-fert | 1.90 | 4.08 | 1.62 | 8.86 [b] | 2.43 | 8.44 | 2.04 | 3.71 [a] | 0.75 | 1.81 |
| 2. Inorganic NPK * | 2.08 | 3.89 | 1.64 | 8.18 [a] | 2.78 | 8.38 | 1.68 | 3.75 [ab] | 0.81 | 1.83 |
| 3. 1.0 t MBM + N₇₉ ** | 2.08 | 4.02 | 1.49 | 8.89 [b] | 2.73 | 8.48 | 2.22 | 4.05 [bc] | 0.88 | 1.90 |
| 4. 1.5 t MBM + N₄₀ *** | 1.89 | 4.09 | 1.47 | 8.94 [b] | 2.61 | 8.52 | 2.21 | 4.01 [abc] | 0.92 | 1.80 |
| 5. 2.0 t MBM **** | 2.06 | 4.02 | 1.56 | 9.08 [b] | 2.81 | 8.56 | 2.23 | 4.15 [c] | 0.79 | 1.98 |
| Mean | 2.00 | 4.02 | 1.56 | 8.79 | 2.67 | 8.47 | 2.08 | 3.94 | 0.83 | 1.86 |

* Inorganic NPK—mineral fertilization; ** MBM + $N_{79}$—meat and bone meal with mineral nitrogen (79 kg N $ha^{-1}$) fertilizers; *** MBM + $N_{40}$—meat and bone meal with mineral nitrogen (40 kg N $ha^{-1}$) fertilizers; **** MBM—meat and bone meal fertilizer. Every year, mineral potassium fertilizer was applied at the same rate of 145 kg K $ha^{-1}$ in treatments 2, 3, 4 and 5; a, b, c, ab, bc, abc—significant differences between means for fertilization (in columns), according to Tukey's test ($p < 0.05$). The absence of superscript letters (in columns) indicates no significant differences between means.

In maize grown for silage, differences in the P content of herbage were not significant. Each species was grown twice in the crop rotation system to facilitate the comparison of results. Significant differences in the P content of seeds of winter oilseed rape were found in the growing season of 2015/16. In this season, the seeds of oilseed rape supplied with mineral fertilizers had significantly lower P content (8.18 g $kg^{-1}$ DM), compared with the remaining treatments (by 8.5% on average), both fertilized and unfertilized. It should be noted that increasing rates of P (45, 68, and 90 kg $ha^{-1}$) were introduced to soil with increasing MBM doses (1.0, 1.5, and 2.0 t $ha^{-1}$). The application of the lowest MBM dose (1.0 t $ha^{-1}$), which supplied 45 kg P $ha^{-1}$, caused a significant (approx. 9%) increase in the P content of seeds of winter oilseed rape, compared with the equivalent rate of P in the form of granular triple superphosphate (20.1% P). The seeds of oilseed rape were more abundant in P in the zero-fert treatment than in the NPK treatment. A lower concentration of a given nutrient in plants may result from the "dilution effect", observed during rapid crop yield growth [18]. The seed yield of winter oilseed rape was highest in the NPK treatment—1.8-fold higher on average than in the zero-fert treatment [15]. A smaller, yet considerable, difference in yields was observed in MBM treatments where the average seed yield of oilseed rape was 1.5-fold higher than in the zero-fert treatment. The P content of seeds of oilseed rape tended to increase with a rise in the MBM dose, but the noted differences were not significant (Table 2). In 2017, the P concentration in seeds of winter oilseed rape were less (by 0.32 g $kg^{-1}$ DM on average) than in 2016. Similarly to 2016, P content was lowest in the seeds of oilseed rape supplied with mineral fertilizers, and highest in those receiving the highest MBM dose, but the observed differences were not significant. The average P content of straw of winter oilseed rape was 2.67 g P $kg^{-1}$ DM in 2016 and 2.08 g P $kg^{-1}$ DM in 2017, and it was not significantly affected by fertilization in either year. The present findings indicate mineral P fertilizers can be replaced with MBM

at the equivalent P rate in the cultivation of winter oilseed rape. According to Nogalska and Zalewska [19], P content in the seeds of winter oilseed rape was higher in response to mineral P fertilizers than MBM, but the noted differences were not significant. In white mustard, only the moderate and high doses of MBM (0.4% and 0.8% MBM per 10 kg of soil in the pot) produced similar effects as mineral fertilizers, whereas the lowest dose (0.2%) significantly reduced the P content of aerial biomass [20].

The grain of winter wheat, grown in 2017/18, was characterized by significant differences in P content which ranged from 3.71 g kg$^{-1}$ DM in the zero-fert treatment to 4.15 g kg$^{-1}$ DM in the 2.0 t ha$^{-1}$ MBM treatment (Table 2). The P content of wheat grain was significantly higher in 1.0 t ha$^{-1}$ MBM and 2.0 t ha$^{-1}$ MBM treatments than in the zero-fert treatment. The grain of winter wheat supplied with mineral fertilizers had significantly lower P content than that supplied with the highest MBM dose. A considerable difference in the P content of wheat straw was found between the years of wheat cultivation—1.56 g P kg$^{-1}$ DM in 2015 and 0.83 g P kg$^{-1}$ DM in 2018. Meat and bone meal had a beneficial influence on the P content of cereal grain in a four-year field experiment [19]. In the cited study, the grain of winter wheat and winter triticale had higher P content after the application of MBM than mineral fertilizers. An increase in the P content of barley grain in response to MBM was reported by Haraldsen et al. [21]. In turn, Nogalska [18] demonstrated that low MBM doses (1.0 and 1.5 t ha$^{-1}$) and mineral fertilizers exerted similar effects on the P content of barley grain, which decreased significantly after the application of high MBM doses (2.0 and 2.5 t ha$^{-1}$). Meat and bone meal can be a rich source of P for oilseed rape, maize, and other cereals and grasses, as well as sugar beets and carrots, both directly after application and in the long term [4,6–8,19,22–30].

Maize was grown for silage in the first (2014) and last (2019) year of the experiment (Table 2). As already mentioned, P levels in maize herbage remained relatively stable in all treatments. The P content of maize herbage was only slightly lower in the zero-fert treatment (1.85 g kg$^{-1}$ DM, mean of two years) than in NPK and MBM treatments (1.96 and 1.95 g P kg$^{-1}$ DM, respectively). The average P content of maize herbage reached 2.0 g kg$^{-1}$ DM in 2014, and it was nearly 0.14 g kg$^{-1}$ lower in 2019. Stępień et al. [9] noted a significant increase (by 33% on average) in the P content of maize grain in the third year of MBM application at 1.0, 2.0, and 3.0 t ha$^{-1}$, relative to the first year. In a study by Nogalska and Zalewska [19], the P content of maize grain increased in response to a high MBM dose (2.5 t ha$^{-1}$), relative to the NPK treatment.

### 3.2. Phosphorus Uptake by Crops

Nutrient uptake is one of the determinants of plant growth and development and, in consequence, the quantity and quality of crop yields. It is estimated based on the produced biomass and the content of the analyzed nutrients in biomass, and constitutes a key criterion for fertilizer evaluation. During the six-year field experiment, P uptake varied depending on fertilizer type and crop species (Table 3).

Phosphorus uptake (on a DM basis, mean of two years) was highest in maize (40 kg ha$^{-1}$), followed by winter wheat and winter oilseed rape (approx. 31 kg ha$^{-1}$ each). Phosphorus uptake by all tested crops was significantly higher in fertilized treatments than in the zero-fert treatment. The only exception was maize harvested in the last year of the study. Phosphorus uptake by maize was 11.5 kg ha$^{-1}$ higher in the NPK treatment than in the zero-fert treatment, but this difference was not significant. It should be stressed that P uptake by the analyzed crop species did not differ significantly between fertilized treatments (including the application of both mineral fertilizers and MBM). This indicates that P from granular triple superphosphate (20.1% P) and MBM was equally efficiently utilized by plants. Nogalska and Zalewska [19] demonstrated that P uptake by winter wheat and maize was higher in MBM treatments than in the NPK treatment. In turn, Brod et al. [22] found no significant differences in P uptake by grasses grown in soil fertilized with mineral fertilizers and MBM. The grain of maize supplied with MBM

every second year at 5.0 t ha$^{-1}$ had significantly higher P content than that in the remaining treatments, and the plants were characterized by the highest P uptake [8].

**Table 3.** The effect of increasing MBM doses on phosphorus uptake (kg ha$^{-1}$ DM) by crops in rotation.

| Treatments | 2014 Silage Maize | 2014/15 Winter Wheat | 2015/16 Winter Oilseed Rape | 2016/17 Winter Oilseed Rape | 2017/18 Winter Wheat | 2019 Silage Maize |
|---|---|---|---|---|---|---|
| 1. Zero-fert | 26.09 [a] | 24.89 [a] | 19.58 [a] | 17.98 [a] | 12.25 [a] | 31.91 [a] |
| 2. Inorganic NPK * | 38.79 [b] | 50.46 [b] | 38.28 [b] | 34.16 [b] | 20.34 [b] | 43.42 [ab] |
| 3. 1.0 t MBM + N$_{79}$ ** | 39.26 [b] | 48.69 [b] | 37.00 [b] | 33.81 [b] | 20.97 [b] | 49.08 [b] |
| 4. 1.5 t MBM + N$_{40}$ *** | 38.65 [b] | 46.77 [b] | 34.95 [b] | 33.07 [b] | 19.29 [b] | 45.01 [b] |
| 5. 2.0 t MBM **** | 39.66 [b] | 45.26 [b] | 31.82 [b] | 28.62 [b] | 19.05 [b] | 47.05 [b] |
| Mean | 36.49 | 43.21 | 32.33 | 29.53 | 18.38 | 43.29 |

* Inorganic NPK—mineral fertilization; ** MBM + N$_{79}$—meat and bone meal with mineral nitrogen (79 kg N ha$^{-1}$) fertilizers; *** MBM + N$_{40}$—meat and bone meal with mineral nitrogen (40 kg N ha$^{-1}$) fertilizers; **** MBM—meat and bone meal fertilizer. Every year, mineral potassium fertilizer was applied at the same rate of 145 kg K ha$^{-1}$ in treatments 2, 3, 4 and 5; a, b, ab—significant differences between means for fertilization (in columns), according to Tukey's test ($p < 0.05$).

In the current study, the greatest variation in P uptake between the years of cultivation was noted in winter wheat, and it was 2.3-fold higher in 2014/15 than in 2017/18 (Table 3). This difference resulted from the fact that winter wheat yield was over two-fold higher in 2015 than in 2018 [15]. Considerable differences in the yield and, consequently, P uptake, were observed in spring barley supplied with MBM during a two-year field experiment [18]. In a pot experiment conducted by Ylivainio et al. [29], P uptake by ryegrass (*Lolium multiflorum*) increased over three years. In the present study, the lower yield of winter wheat in 2018 resulted from adverse weather conditions during the growing season of 2017/2018 (Table 3). In maize and winter oilseed rape, the differences in P uptake between the years of cultivation were small and reached 6.8 and 2.8 kg ha$^{-1}$ on average, respectively.

### 3.3. Soil pH

The pH in 1 M KCl dm$^{-3}$ of soil (mean of six years) ranged from 5.24 in the NPK treatment to 5.36 in the treatment with the highest MBM dose without supplemental mineral N (Table 4).

**Table 4.** The effect of increasing MBM doses on the pH$_{KCl}$ of soil and the available phosphorus content (mg kg$^{-1}$ DM) of soil, mean for 2014–2019.

| Treatment | | pH$_{KCL}$ | P |
|---|---|---|---|
| 1. Zero-fert | | 5.35 | 69.01 [a] |
| 2. Inorganic NPK * | | 5.24 | 70.79 [a] |
| 3. 1.0 t MBM + N$_{79}$ ** | | 5.27 | 72.98 [a] |
| 4. 1.5 t MBM + N$_{40}$ *** | | 5.35 | 82.79 [ab] |
| 5. 2.0 t MBM **** | | 5.36 | 90.92 [b] |
| Annual mean | 2014 | 5.40 [C] | 79.26 [C] |
| | 2015 | 5.10 [A] | 68.06 [AB] |
| | 2016 | 5.38 [BC] | 60.48 [A] |
| | 2017 | 5.33 [BC] | 89.08 [D] |
| | 2018 | 5.27 [B] | 75.66 [BC] |
| | 2019 | 5.40 [C] | 91.23 [D] |
| Interaction (f × y) | | ns | s |

* Inorganic NPK—mineral fertilization; ** MBM + N$_{79}$—meat and bone meal with mineral nitrogen (79 kg N ha$^{-1}$) fertilizers; *** MBM + N$_{40}$—meat and bone meal with mineral nitrogen (40 kg N ha$^{-1}$) fertilizers; **** MBM—meat and bone meal fertilizer. Every year, mineral potassium fertilizer was applied at the same rate of 145 kg K ha$^{-1}$ in treatments 2, 3, 4 and 5; a, b, ab—significant differences between means for fertilization (in columns), A, B, C, D, AB, BC—significant differences between means for the years 2014–2019 (in column), according to Tukey's test ($p < 0.05$). The absence of superscript letters (in columns) indicates no significant differences between means. Interaction between fertilization and year (f × y); s—significant; ns—not significant.

Fertilization had no significant effect on soil pH. At the beginning of the experiment, the soil was slightly acidic (pH$_{KCl}$ = 5.61), but its pH decreased during the six-year crop cultivation in both NPK and MBM treatments, and it was classified as acidic at the end of the study. Higher MBM doses (1.5 and 2.0 t ha$^{-1}$) supplied considerable quantities of Ca (from 150 to 200 kg ha$^{-1}$), which increased soil pH (relative to the NPK treatment), but the noted values were similar to that in the zero-fert treatment. The MBM used in this experiment was slightly acidic (pH$_{H2O}$ = 6.3). A minor increase in soil pH in response to the application of MBM was also reported by Valenzuela et al. [31] and Deydier et al. [32]. In a pot experiment where white mustard was grown in soil fertilized with MBM, pH increased considerably and changed from acidic to slightly acidic [20]. In other studies, soil pH decreased with increasing MBM doses [20,26,30], which indicates that the acidifying effect of ammonia released from MBM was stronger than the buffering effect of Ca [19].

The values of soil pH varied significantly across the years of the study, however the results from particular years are difficult to interpret. Soil pH was highest in the first and last year of the experiment (pH$_{KCl}$ = 5.40), which suggests that MBM (in particular at higher doses) stabilizes soil pH. This indicator could also be affected by crop species. The highest pH values were observed in soil under maize—they were significantly higher than in the second and fifth year of the study when winter wheat was grown. After the harvest of winter wheat, soil had the lowest pH values which differed significantly between the years of its cultivation; the average pH$_{KCl}$ was 5.10 and 5.27 in 2015 and 2018, respectively (the soil remained acidic). The significant differences in the values of soil pH between the growing seasons of winter wheat were due to uneven precipitation and considerable differences in nutrient uptake by plants. After the harvest of winter oilseed rape, soil samples collected in the third and fourth year of cultivation had a similar pH (pH$_{KCl}$ = 5.38 and 5.33). Soil pH decreased significantly after the harvest of winter wheat (second year of the study). The results of previous studies are inconclusive and often contradictory due to the presence of various factors such as soil type, weather conditions, duration of the experiment, and the type and dose of animal waste [22,31,33]. Stępień and Mercik [34] observed a decrease in the pH of soil amended with MBM, horn meal, and feather meal, whereas Spychaj-Fabisiak et al. [35] found no differences in this parameter after the application of a conditioner based on poultry offal. Soil acidification may be affected by the activity of nitrifying microorganisms whose growth is promoted by the application of organic wastes and under high sulfate concentrations [33].

### 3.4. Available Phosphorus Content of Soil

In the six-year crop rotation system, the average available P content of soil increased from the initial value of 65.0 to 82.2 mg P kg$^{-1}$ (26% increase) after the application of MBM, and to 70.8 mg P kg$^{-1}$ (approx. 9% increase) after the application of mineral fertilizers (Table 4). The available P content of soil increased with a rise in the MBM dose, but a significant increase (by 32%, over 28%, and 25% relative to treatments No. 1, 2, and 3, respectively) was noted only in response to the highest MBM dose which supplied 90 kg P ha$^{-1}$. Simoes et al. [27] also noted a close correlation between MBM dose and the available P content of soil. Lower MBM doses (1.0 and 1.5 t ha$^{-1}$) increased P abundance in soil by around 9%, whereas higher MBM doses (2.0 and 2.5 t ha$^{-1}$) increased it by 18% and 25%, respectively, compared with mineral fertilizer [26]. A beneficial influence of MBM on the available P content of soil has been reported by numerous authors [22,29–31,33]. The present study demonstrated that the lowest rate of P (45 kg ha$^{-1}$) in the form of MBM can replace mineral fertilizer at the equivalent rate.

The average available P content of soil varied significantly across years of the study (Table 4), and it was highest in the sixth (last) year (91.23 mg kg$^{-1}$ DM) and in the fourth year (89.08 mg kg$^{-1}$ DM); the noted increase was significant relative to the remaining experimental years. The average P uptake by maize, which was grown in the last year of the study, exceeded 43 kg ha$^{-1}$, whereas the average P uptake by winter oilseed rape, which was grown in the fourth year, was 14 kg lower (Table 3). It appears that such a high

content of available P in soil in the last year was due to its accumulation in soil, resulting from the supply of large quantities of P (90 kg ha$^{-1}$) with MBM applied at 2.0 t ha$^{-1}$ each year. However, a steady increase in soil P abundance, reported previously [26,30], was not observed because the available P content of soil was lowest in the third year of the experiment. According to many authors [6,19,31,36], if P availability is low during the first growing season, MBM can become the main source of this nutrient for crop plants in subsequent years, and P fertilizers should not be applied in the following year. The amount of P available to plants is determined by soil pH, moisture content, and organic matter content. In the present study, the abundance of available P in soil increased with a rise in soil pH in MBM treatments (Table 4). The opposite trend was noted in experiments conducted by Nogalska and Zalewska [19] and Nogalska et al. [26] where the available P content of soil increased with a drop in pH. However, soil pH was considerably higher in the cited studies (pH$_{KCl}$ = 6.5 on average (slightly acidic)) than in the current experiment (pH$_{KCl}$ = 5.3 (acidic)). Slightly acidic soils are most abundant in available P. In MBM, P is present in organic form (meat fraction) and in the form of hydroxyapatite (bone fraction). Organic P is readily available to plants, whereas P from hydroxyapatite is released in an acidic environment [6] and in soils colonized by mycorrhizal fungi [5] and/or P solubilizing bacteria [11,20,37]. The availability and mobility of P are determined not only by the physicochemical and mineralogical properties of bone apatite [38], but also by environmental and biological factors. A mechanistic understanding of the temporal transformation and aging of bones in the field can provide new insights into the sustainable cycling of P. In comparison with rock apatite, bone apatite and hydroxyapatite can be alternative sources of efficient, easily accessible, and economical P fertilizer [39,40].

### 3.5. Soil Phosphorus Balance

During the six-year experiment, the following quantities of P were introduced to soil: 270 kg ha$^{-1}$ in treatments No. 2 and 3, 408 kg ha$^{-1}$ in treatment No. 4, and 540 kg ha$^{-1}$ in treatment No. 5 (Table 5).

**Table 5.** Calculated cumulative phosphorus balance for the years 2014–2019.

| Treatment | Dose (kg ha$^{-1}$) | Uptake (kg ha$^{-1}$) | Balance (kg ha$^{-1}$) | Utilization (%) |
|---|---|---|---|---|
| 1. Zero-fert | 0 | 132.6 | −132.6 | |
| 2. Inorganic NPK * | 270 | 224.9 | 45.1 | 34.2 |
| 3. 1.0 t MBM + N$_{79}$ ** | 270 | 228.8 | 41.2 | 35.6 |
| 4. 1.5 t MBM + N$_{40}$ *** | 408 | 218.7 | 189.3 | 21.1 |
| 5. 2.0 t MBM **** | 540 | 211.5 | 328.5 | 14.6 |
| Mean | 372 | 203.3 | 94.3 | 26.4 |

* Inorganic NPK—mineral fertilization; ** MBM + N$_{79}$—meat and bone meal with mineral nitrogen (79 kg N ha$^{-1}$) fertilizers; *** MBM + N$_{40}$—meat and bone meal with mineral nitrogen (40 kg N ha$^{-1}$) fertilizers; **** MBM—meat and bone meal fertilizer. Every year, mineral potassium fertilizer was applied at the same rate of 145 kg K ha$^{-1}$ in treatments 2, 3, 4 and 5.

Total P uptake by plants ranged from 132.6 kg ha$^{-1}$ in the zero-fert treatment to 228.8 kg ha$^{-1}$ in the treatment with the lowest MBM dose (1.0 t ha$^{-1}$) (Table 3). It should be noted that P uptake by plants decreased gradually from 228.8 to 211.5 kg ha$^{-1}$, and P utilization decreased from 35.6% to 14.6% with increasing P rates (Table 5). Phosphorus supplied by the lowest dose of MBM was most efficiently utilized by plants (in approx. 36%), and P uptake and balance in this treatment were comparable with those in the NPK treatment. In both treatments, the annual P rate was 45 kg ha$^{-1}$, which was sufficient to meet the P requirements of plants. This indicates that P from MBM and granular triple superphosphate was equally efficiently utilized by the tested crops. High utilization of P from MBM by maize was also observed by Nogalska et al. [8] who found that P uptake was higher after the application of MBM than granular triple superphosphate (46%). In the present study, MBM applied at 1.5 and 2.0 t ha$^{-1}$, which each year supplied 68 and 90 kg P ha$^{-1}$, respectively, contributed to considerable accumulation of P in soil—189

and 328 kg ha$^{-1}$, respectively. Therefore, the question arises whether P that has not been utilized, and accumulates in soil after the application of MBM, can become available to plants. In this field experiment, the abundance of available P in soil did not increase steadily over six years (Table 4). However, its significant increase was noted in the fourth and sixth (last) years, which indicates that the available P content of soil is affected by factors such as crop species, weather conditions, and soil pH. It should also be stressed that the analyzed soil was slightly acidic at the beginning and acidic at the end of the experiment. Bone apatite provides a slow and steady dissolution of P after soil application, in comparison with rock mineral fertilizer [41]. In acidic soils, sparingly soluble P compounds from bone hydroxyapatite can be relatively well mobilized, thus improving P availability to plants. Therefore, the application of high MBM doses can lead to the accumulation of large amounts of P, in particular in acidic soils [42]. In a study by Ylivainio et al. [29], 5–28% of P was available to ryegrass in the first year after MBM application, followed by 63% in the second year and 69–87% in the third year. According to Jeng et al. [6], as much as 50% of P from MBM is available to plants in the first year. The results of other studies also show that MBM can be a rich source of available P already in the first year after its application [8,30]. In the current study, soil was moderately abundant in available P; therefore, plants could absorb P from soil reserves, which partially met their requirements. The fertilizing effect of MBM is generally greater in soils characterized by low abundance of available P [22,25].

## 4. Conclusions

Meat and bone meal, a by-product of the meat industry, is an important pathway for N and P recycling, which contributes to sustainable nutrient management in line with the European Green Deal. This strategy promotes lower energy use in mineral fertilizer production, environmental protection, and circular economy. The six-year field experiment revealed that MBM is a rich source of P for silage maize, winter wheat, and winter oilseed rape. The lowest dose of MBM (1.0 t ha$^{-1}$), which supplied 45 kg P ha$^{-1}$, was sufficient to meet the P requirements of crops to the same extent as the equivalent rate of P in the form of granular triple superphosphate (20.1% P). The uptake, balance, and utilization of P by plants were comparable in both treatments. In turn, two higher MBM doses (1.5 and 2.0 t ha$^{-1}$) applied each year during the six-year crop rotation system contributed to considerable accumulation of available P in soil (189 and 328 kg P ha$^{-1}$, respectively). Soil pH was not significantly affected by MBM. MBM can replace conventional mineral P fertilizers in crop cultivation. However, MBM should not be applied at doses exceeding 1.5 t ha$^{-1}$ to crops grown in acidic soils where sparingly soluble P compounds are relatively easily converted into the water-soluble form. The optimal dose of MBM should be established based on P uptake by a given crop species.

**Author Contributions:** Conceptualization, A.N. and A.Z.; methodology, A.N. and A.Z.; chemical analyses, A.Z.; data analysis, A.N. and A.Z.; investigation, A.N. and A.Z.; resources, A.N. and A.Z.; data curation, A.N. and A.Z.; writing, A.Z.; writing—review and editing, A.N.; visualization, A.Z.; supervision, A.N.; funding acquisition, A.N. All authors have read and agreed to the published version of the manuscript.

**Funding:** This research was supported by the Ministry of Education and Science as part of statutory activities (No. 30.610.003-110). Project financially supported by Minister of Education and Science in the range of the program entitled "Regional Initiative of Excellence" for the years 2019–2022, Project No. 010/RID/2018/19, amount of funding 12.000.000 PLN.

**Institutional Review Board Statement:** Not applicable.

**Informed Consent Statement:** Not applicable.

**Data Availability Statement:** Not applicable.

**Conflicts of Interest:** The authors declare no conflict of interest.

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
