# Peer review of "The Effect of Meat and Bone Meal (MBM) on Phosphorus (P) Content and Uptake by Crops, and Soil Available P Balance in a Six-Year Field Experiment"

_sustainability, doi:10.3390/su14052855_

Round 1

Reviewer 1 Report

 The  manuscript "The Effect of Meat and Bone Meal (MBM) on Phosphorus (P) Content and Uptake by Crops, and Soil Available P Balance in 3
a Six-Year Field Experiment: is well structured and written. The following things nee to be improved.  Abstract sections are too concise need to elaborate more with proper gist. n K (163 mg K kg-1) check the unit as well.  ## Latest Citation is missing. The paper is well written could be published.

Author Response

Thank you very much for the time and effort devoted to reviewing the manuscript. A long-term (6-year) field experiment involving three crop species provided a huge amount of data. Due to the limited word count in the abstract, the authors focused on describing the main methodological assumptions and writing the main conclusions. The unit (163 mg K kg-1) was checked and is correct. The paper was completed in early January 2022, so it was not possible to include the most recent research. It is worth noting that studies from 2021 (citations number 2, 9, 11, and 15) and 2020 (citations number 5, 13, 14, 38) were included in the paper. Thank you for all your comments. 

Reviewer 2 Report

Fine study, with excellent data processing and discussion, having importance from the scientific and practical point of view it the term of waste recycling in the form of fertilizer.

There are some typing and style errors present through the text, requiring carefully reading and correction.

Since authors have already published one part of the results achieved in greater experimental trial with MBM incorporation (Ref. 15), and they comment on data present in Ref. 15 as a part of this study, it would be important to the potential readers to underline this fact, i.e. if it is hard to change title, than in aim, as well as in M&M section it should be repeated/emphasized.

Please use SI units in manuscript, instead of Mg use t (ton).

Line 70: Was the meat and bone meal processed previous to the incorporation into soil? If the answer is positive, please include short description, if it is possible.

Line 107-111: Please include sowing, as well as harvesting dates for each crop.

Line 122-123: Please, include reference for applied analytical procedure, if it is possible.

Figure 2: Please, switch columns into lines, it will be clearer to readers to see temperature variations across seasons.

In R&D section it will be better to avoid phrase such as: "the seeds were less abundant in P", and instead to use phrase such as: the P concentration in seeds.... as proper explanation.

Line 214: meat and bone meat - please correct to meat and bone meal.

Applied crop rotation is little bit unusual, the sequences are not repeating across years. Accordingly, it is clear that P outtake by crops and P accumulation in soil is depending on crop (including species and genotypic P using efficiency - PUE), as well as meteorological conditions, while the impact of rotation is present, but hard to explain and understand. It seems that soil microbiota play an important role in MBM decomposition, particularly from the bone hydoxyapatite. Greater crop efficacy, driven by lower MBM doses indices that MBM could not be incorporated each year and advantage should be given to smaller doses. Thus, MBM could serve as an excellent P source, un-affecting other soil characteristics.

I could encourage authors to continue research regarding soil microbiota and additional increase of soil fertility.

Author Response

R. Fine study, with excellent data processing and discussion, having importance from the scientific and practical point of view it the term of waste recycling in the form of fertilizer.

A. Thank you very much for the time and effort devoted to reviewing the manuscript. MBM, a by-product of the meat industry, is an important pathway for N and P recycling in sustainable nutrient cycling and organic matter management, in line with the European Green Deal that promotes lower energy use in mineral fertilizer production, environmental protection and circular economy. All of the reviewer's comments of the paper have been addressed, and the responses to those comments can be found below.

Other Reviewer’s comments:

R. There are some typing and style errors present through the text, requiring carefully reading and correction.

A. Thank you for pointing this out. The manuscript has been carefully read again with attention to errors.

R. Since authors have already published one part of the results achieved in greater experimental trial with MBM incorporation (Ref. 15), and they comment on data present in Ref. 15 as a part of this study, it would be important to the potential readers to underline this fact, i.e. if it is hard to change title, than in aim, as well as in M&M section it should be repeated/emphasized.

A. Previously published results from the experiment are emphasized in the aim of the paper, and citations of relevant publications (citation number 13, 14, and 15). Also in the M&M section, an article containing data from a six-year experiment conducted was cited (citation number 15).

R. Please use SI units in manuscript, instead of Mg use t (ton).

A. Thank you for pointing out the units used. All Mg units have been corrected to conform to SI standards (ton – “t”).

R. Line 70: Was the meat and bone meal processed previous to the incorporation into soil? If the answer is positive, please include short description, if it is possible.

A. The meal used in the experiment was not previously processed. A sentence was added to clarify this point: "The meat and bone meal was applied in a loose form and mixed with soil without additional processing".

R. Line 107-111: Please include sowing, as well as harvesting dates for each crop.

A. Sowing and harvest dates are added for each crop grown.

R. Line 122-123: Please, include reference for applied analytical procedure, if it is possible.

A. The vanadium-molybdenum method used in the analysis is widely known and used. It is designated by the Polish Standard BN-81/0520-15.

R. Figure 2: Please, switch columns into lines, it will be clearer to readers to see temperature variations across seasons.

A. The column chart has been corrected to a line chart. Thank you for pointing out that the line chart is clearer to the reader.

R. In R&D section it will be better to avoid phrase such as: "the seeds were less abundant in P", and instead to use phrase such as: the P concentration in seeds.... as proper explanation.

A. Thank you for this comment. The R&D section has been re-read. Reworded the phrase in places where possible.

R. Line 214: meat and bone meat - please correct to meat and bone meal.

A. “meat” was replaced with the correct term “meal” – we apologize for this obvious typing error.

R. Applied crop rotation is little bit unusual, the sequences are not repeating across years. Accordingly, it is clear that P outtake by crops and P accumulation in soil is depending on crop (including species and genotypic P using efficiency - PUE), as well as meteorological conditions, while the impact of rotation is present, but hard to explain and understand. It seems that soil microbiota play an important role in MBM decomposition, particularly from the bone hydoxyapatite. Greater crop efficacy, driven by lower MBM doses indices that MBM could not be incorporated each year and advantage should be given to smaller doses. Thus, MBM could serve as an excellent P source, un-affecting other soil characteristics.

I could encourage authors to continue research regarding soil microbiota and additional increase of soil fertility.

A. Thank you for highlighting the issues of crop rotation and soil microbiota. The authors are currently focusing on data compilation and publication of results from the experiment. A long-term (6-year) field experiment involving three crop species provided a huge amount of data. Thank you very much for suggesting the possibility of continuing the research with soil microbiota. It is a very interesting idea.

Reviewer 3 Report

A well-written paper.

Lines 115-116: How did you collect the samples from each plot? Did you have guard rows between treatment plots?

Line 203: Do you mean “but” instead of “nut”? “…nut the noted differences…”

Author Response

R. A well-written paper.

A. Thank you very much for the time and effort devoted to reviewing the manuscript, and for the positive feedback.

R. Lines 115-116: How did you collect the samples from each plot? Did you have guard rows between treatment plots?

A. Thank you for bringing these issues to our attention. “The crop was harvested with a harvester from each plot. An average representative plant sample of 1.0 kg was taken from each plot. A 1 m guard rows was maintained between the plots." This has been corrected in the manuscript, line 126-127 and 128-130.

R. Line 203: Do you mean “but” instead of “nut”? “…nut the noted differences…”

A. “nut” was replaced with the correct term “but” – we apologize for this obvious typing error.
